# Reproducibility of dynamic contrast enhanced MRI derived transfer coefficient $K^{trans}$ in lung cancer

**Jean-Philip Daniel Weber** [1] *, **Judith Eva Spiro** [2,3], **Matthias Scheffler** [4], **Jürgen Wolf** [4], **Lucia Nogova** [4], **Marc Tittgemeyer** [5], **David Maintz** [1], **Hendrik Laue** [6], **Thorsten Persigehl** [1]

**1** Department of Radiology, University Hospital Cologne, Cologne, Germany, **2** Department of Radiology, University Hospital, Ludwig-Maximilians-Universität München, Munich, Germany, **3** Comprehensive Pneumology Center, Member of the German Center for Lung Research, Munich, Germany, **4** Lung Cancer Group, Department I of Internal Medicine, University Hospital Cologne, Cologne, Germany, **5** Max Planck Institute for Metabolism Research, Cologne, Germany, **6** Fraunhofer Institute for Digital Medicine MEVIS, Bremen, Germany

* jean-philip.weber@uk-koeln.de

**Data Availability Statement:** All relevant data are within the manuscript and its Supporting Information files.

## Abstract

Dynamic contrast enhanced MRI (DCE-MRI) is a useful method to monitor therapy assessment in malignancies but must be reliable and comparable for successful clinical use. The aim of this study was to evaluate the inter- and intrarater reproducibility of DCE-MRI in lung cancer. At this IRB approved single centre study 40 patients with lung cancer underwent up to 5 sequential DCE-MRI examinations. DCE-MRI were performed using a 3.0T system. The volume transfer constant $K^{trans}$ was assessed by three readers using the two-compartment Tofts model. Inter- and intrarater reliability and agreement was calculated by wCV, ICC and their 95% confident intervals. DCE-MRI allowed a quantitative measurement of $K^{trans}$ in 107 tumors where 91 were primary carcinomas or intrapulmonary metastases and 16 were extrapulmonary metastases. $K^{trans}$ showed moderate to good interrater reliability in overall measurements (ICC 0.716–0.841; wCV 30.3–38.4%). $K^{trans}$ in pulmonary lesions ≥ 3 cm showed a good to excellent reliability (ICC 0.773–0.907; wCV 23.0–29.4%) compared to pulmonary lesions < 3 cm showing a moderate to good reliability (ICC 0.710–0.889; wCV 31.6–48.7%). $K^{trans}$ in intrapulmonary lesions showed a good reliability (ICC 0.761–0.873; wCV 28.9–37.5%) compared to extrapulmonary lesions with a poor to moderate reliability (ICC 0.018–0.680; wCV 28.1–51.8%). The overall intrarater agreement was moderate to good (ICC 0.607–0.795; wCV 24.6–30.4%). With $K^{trans}$, DCE MRI offers a reliable quantitative biomarker for early non-invasive therapy assessment in lung cancer patients, but with a coefficient of variation of up to 48.7% in smaller lung lesions.

## Introduction

With the increasing introduction of target specific tyrosine kinase inhibitor and antiangiogenic treatment option for lung cancer, the aspect of quantitative imaging in terms of tumor characterization and treatment monitoring becomes more and more important [1]. Dynamic

**Funding:** The authors received no specific funding for this work.

**Competing interests:** The authors have declared that no competing interests exist.

contrast-enhanced MRI (DCE-MRI) is a frequently used, non-invasive, quantitative modality for analysing tumor vascularisation and (micro-)perfusion without the use of ionizing radiation. DCE-MRI is typically based on measuring the T1-contrast enhancement before, during and after the application of an extracellular gadolinium-based contrast agent [2].

There are a broad number of semiquantitative, and quantitative pharmacokinetic parameters published allowing data analyses in DCE-MRI. Common models are the Tofts-, Brix- and the 2-compartment-exchange model. The most frequently used Tofts-model (also called Generalized Kinetic Model) calculates the transfer constant/permeability surface product ($K^{trans}$) and the flux rate constant ($K_{ep}$) as surrogate parameter of the permeability. It is calculated by optimizing the model parameters to the measured contrast agent concentration curve using the Levenberg-Marquardt algorithm [3]. $K^{trans}$ is a parameter that reflects vascular microperfusion and endothelial permeability in a characterized tissue. Several studies have shown, that $K^{trans}$ is an important parameter for non-invasive characterisation of tumor subtypes and functions as well as a sensitive and early biomarker correlating with treatment response [4]. The RSNA Quantitative Imaging Biomarkers Alliance (QIBA) DCE-MRI Technical Committee released a DCE-MRI Profile with the goal of better standardization of DCE-MRI acquisitions at 1.5T [5]. The committee proposed a within-subject coefficient of variation of 20% for $K^{trans}$, based on a conservative estimate from the peer-reviewed literature outside the lung. Recent studies showed a high reliability of $K^{trans}$ for example in brain gliomas, in solid tumors in children, in liver tumors and in renal cell carcinomas [6–9]. Nevertheless, it is not known whether this reproducibility is also applicable to lung tumors, which are known to be vulnerable to respiratory motion. Inter-frame misalignment of focal lesions makes a tedious frame-by-frame measurement by the radiologist indispensable [10].

The purpose of this study was to analyse the inter- and intrarater reproducibility (ICC) and the within-subject coefficient of variation (wCV) of the pharmacokinetic biomarker $K^{trans}$ in DCE-MRI of lung cancer.

## Materials and methods

### Patient population

MRI data were available from the MIMEB trial (Molecular Imaging and Molecular Markers in NSCLC treated with Erlotinib and Bevacizumab) conducted at the University Hospital of Cologne, Germany [11]. One secondary objective of this trial was the feasibility and reproducibility of DCE-MRI in advanced (stage IV) NSCLC. In this prospective, institutional review board (IRB) approved clinical pilot trial 42 participants with non-small cell lung cancer (NSCLC) were enrolled. Out of these 42 participants, 2 refused the MRI. Thus, 40 participants (23 men) were included. Mean age was 59 ±12 years. Written informed consent was obtained for all participants.

Inclusion criteria was a histologically confirmed non-squamous NSCLC stage IV. Minimum age was ≥18 years. Subjects showed at least one measurable lesion in CT or MRI according to Response Evaluation Criteria in Solid Tumors (RECIST). Creatinine clearance conducted within 7 days prior to DCE-MRI examination had to be ≥60 ml/min. Exclusion criteria from MRI were metallic implants, claustrophobia or known allergic reaction to gadolinium. Criteria for discontinuation were voluntary discontinuation, severe non-compliance, patient lost to follow-up, disease progression, unacceptable toxicity, death, or pregnancy.

After baseline examination all patients were treated with Erlotinib and Bevacizumab during a period of six weeks. Follow-up DCE-MRI was performed in week 2 and 7. In cases of non-progression, the following MRI examinations were performed every six weeks.

## MR imaging

All DCE-MRI examinations were performed using a 3.0T MRI system (Trio Tim, Siemens AG, Medical Solutions). It has a gradient system with 40 mT/m maximum amplitude and a slew rate of 200 T/m/s. For tumor localization, coronal and transversal T2-weighted single-shot turbo spin-echo (HASTE) sequences were used (resolution matrix: 320 x 320; number of slices: 35; field of view: 450 x 450 mm; slice thickness: 5 mm; slice gap: 1 mm; repetition time: 2000 ms; echo time: 92 ms; flip angle: 180˚). To determine the T1 relaxation time in blood vessels and tumor tissue, two pre-contrast T1-weighted volumetric interpolated breath-hold examination (VIBE) sequences with different flip angles were used (flip angles: 2˚ and 15˚). For dynamic contrast enhancement (DCE) measurements, a free breathing T1-weighetd time-resolved angiography with stochastic trajectories (TWIST) was acquired over 9 min with 86 phases and scan time of 6.23 s per phase (resolution matrix: 192 x 104; number of slices: 35; field of view: 380 x 285 mm; slice thickness: 4 mm; repetition time: 4.44 ms; echo time: 1.72 ms; flip angle: 12˚). The first 5 measurements before injection of the contrast agent were used to create a reliable baseline for DCE analysis. For the DCE examination, the extracellular contrast agent gadopentetate dimeglumine (Gd-DTPA; Bayer Healthcare, Berlin, Germany) was administered according to the individual patient body weight (0.5 mmol/kg per body weight). The contrast agent was administered via a catheter (20 gauge or 22 gauge) in the cubital vein with a constant flow of 2 ml/s followed by 20 ml saline flush using an automatic injection system. All mentioned acquisition process steps are subject to quality control steps as outlined in QIBA-profile v1.0 [5].

## DCE-MRI analysis

All measurements were performed by three radiologists with an experience in MRI of over 3 years, 5 years and 15 years. The localization of the index lesions was annotated by the most experienced radiologist and presented to the other readers with an arrow. To determine the pharmacokinetic parameters, several manual ROIs were placed in the tumor slice by slice covering the whole tumor. The number of slices, the exact placement and the size of the ROIs were freely and individually determined by each reader. The only requirement was not to exceed the outer tumor margins. Slice positions and imaging phases with strong artefacts within the dynamic sequence were excluded. For intraclass correlation, the measurements were repeated by two of the above-mentioned radiologists after a time interval of one month. Both readers were blinded to the initial results.

Using the dedicated DCE-MRI software PulmoMR (Mevis Fraunhofer, Bremen, Germany), the pharmacokinetic parameter $K^{trans}$ for each ROI was computed by using the two-compartment Tofts-model and the population average arterial input function (AIF) of Weinmann.

## Statistical analysis

Mean $K^{trans}$ value of every lesion was measured. To determine inter- and intraobserver agreement, the Intraclass Correlation Coefficient (ICC) and its 95% confident intervals (CI) were calculated using SPSS statistical package version 25 (SPSS Inc, Chicago, IL). ICC was calculated based on a single-measurement, absolute-agreement, and 2-way mixed-effects model. Values less than 0.50 indicate poor reliability, values between 0.50 and 0.75 indicate moderate reliability, values between 0.75 and 0.90 indicate good reliability, and values greater than 0.90 indicate excellent reliability [12]. To determine the variability of $K^{trans}$, the within-subject coefficient of variation (wCV) and its 95% confident intervals (CI) were calculated based on a cross-sectional claim using the root mean square approach.

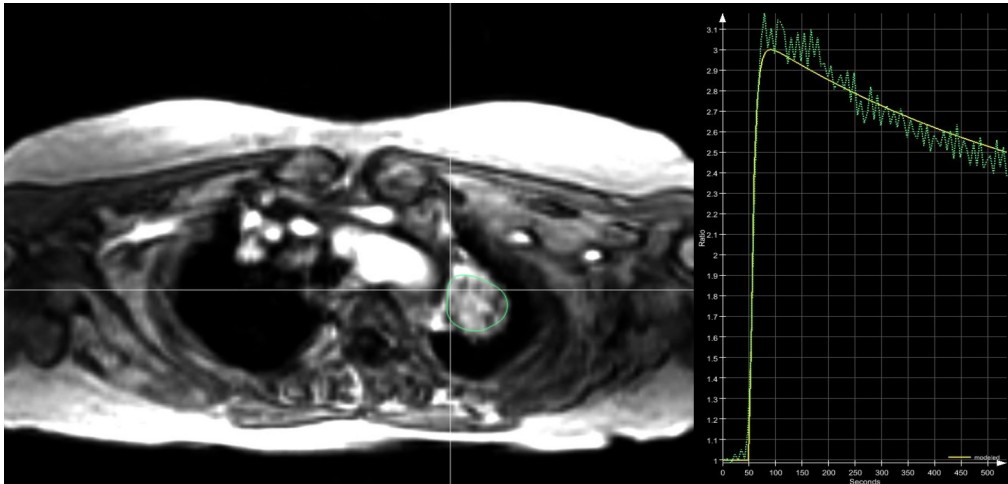

**Fig 1. ROI placement and K^trans analysis.** Example of ROI placement in a lung tumor of the left upper lobe (left) and derived time curve of contrast agent enhancement (right) using the PulmoMR software (Mevis Fraunhofer, Bremen, Germany).

## Results

### MRI imaging

Within the 40 patients up to 5 follow-up examinations during a period of up to 27 months were analysed. In most patients one index lesion was detectable, in 3 patients two different tumor lesions were measured. In 1 patient baseline DCE-MR imaging was performed twice in an interval of 7 days. Including all baseline and follow-up examinations, this leads to a total amount of 128 tumor lesions in 40 patients being enrolled. However, 21 lesions were excluded from the analyses due to missing imaging data or inadequate DCE image quality, leading to postprocessing errors (n = 8), poor visualisation of the tumor (n = 7) and very small lesions < 1 cm (n = 6). Thus, $K^{trans}$ was measured in 107 tumors at 40 patients using the DCE-MRI software PulmoMR (Fig 1).

For DCE-MRI analysis the tumors were subdivided in clinical location and size (Table 1). In total, 91 tumors were intrapulmonary lesions including primary lung cancer as well as intra-pulmonary metastases and relapsed bronchial carcinomas. Among the intrapulmonary lesions 50 lesions were ≥ 3 cm and 41 lesions < 3 cm. The mean diameter of the intrapulmonary lesions was 3.8 ± 2.3 cm. In addition, 16 extrapulmonary lesions were analysed including 10 mediastinal lymph nodes, 6 located paratracheal and 4 located subcarinal. The mean diameter

**Table 1. Patient number and lesion characteristics.**

| | | |
|---|---|---|
| **Number of patients** | | 40 |
| **Number of lesions** | | 107 |
| **Location intrapulmonary** | 1–3 cm | 41 |
| | >3 cm | 50 |
| **Location extrapulmonary** | Bone | 6 |
| | Lymph node | 10 |
| **Major Responder** | | 11 |
| **Minor Responder** | | 24 |

Overview of patient and lesion number depending on their size and location.

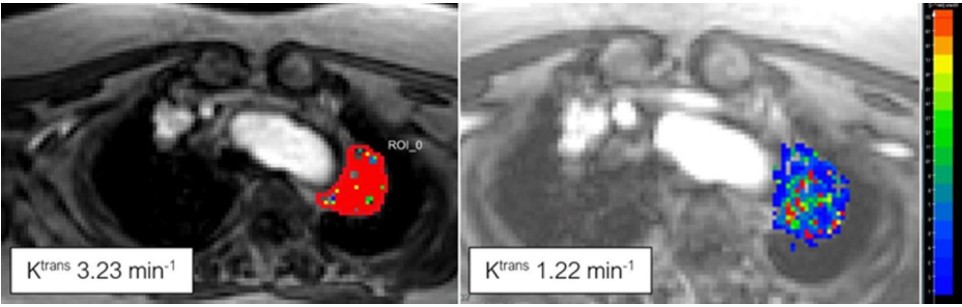

**Fig 2. Example of K^trans decrease during therapy.** NSCLC of the left upper lobe showing a clear decrease of K$^{trans}$ from baseline (left) to first follow-up after 1 week of checkpoint inhibition by Erlotinib and Bevacizumab.

of the lymph nodes was 2.6 ± 0.5 cm. Moreover, of the 16 extrapulmonary tumors 6 lesions were osteolytic bone metastases within the ribs with a soft tissue component and a mean diameter of 4.3 ± 0.8 cm.

DCE-MRI allowed a quantitative measurement of K$^{trans}$ at baseline and follow-up (Fig 2). Overall, the median value of K$^{trans}$ was 0.53 min$^{-1}$ with a range above all measurements of 0.02 to 6.64 min$^{-1}$ while 74% of the measurements revealed a K$^{trans}$ value of less than 1.0 min$^{-1}$. Intrapulmonary lesions demonstrated a median K$^{trans}$ value at baseline of 0.79 min$^{-1}$, first follow-up of 0.43 min$^{-1}$, second follow-up of 0.33 min$^{-1}$, third follow-up of 0.35 min$^{-1}$. The detailed K$^{trans}$ values during the tumor therapy for the different tumor locations are shown in Table 2 and Fig 3.

Moreover, all patients were categorized in major and minor responders depending on their relative percentage change of tumor size in the last available follow-up compared to the baseline examination in line to the RECIST 1.1. Major responders (n = 11) with a partial response demonstrated a decrease of mean K$^{trans}$ from baseline of 0.79 min$^{-1}$ to first follow-up of 0.47 min$^{-1}$, minor responder (n = 24) with stable disease of 0.70 ± 0.95 to 0.45 ± 0.52 without reaching statistically significant difference between the response groups. However, there was a trend towards a higher decrease in the partial response compared to the stable disease group as displayed in Fig 4; detailed changes of lesion sizes and corresponding K$^{trans}$ are shown in Fig 4 and Table 3. Note that Patients who underwent only the baseline MR examination (n = 3) and showed only extrapulmonary lesions (n = 2) were excluded in this figure. At this point it should be noted that the therapeutic effect on the K$^{trans}$ values is not the goal of this study, but merely serves to better visualize the K$^{trans}$ function and to illustrate its potential clinical benefit.

**Table 2. K^trans values.**

| | Baseline | Follow-up 1 | Follow-up 2 | Follow-up 3 |
|---|---|---|---|---|
| **Overall** | 0.80 (0.48; 1.42) | 0.44 (0.24; 0.87) | 0.34 (0.17; 0.71) | 0.38 (0.29; 0.68) |
| **Pulmonary** | 0.79 (0.46; 1.38) | 0.43 (0.24; 0.82) | 0.33 (0.16; 0.59) | 0.35 (0.29; 0.52) |
| **Minor Response** | 0.70 (0.51; 1.34) | 0.45 (0.26; 0.81) | 0.43 (0.30; 0.61) | 0.34 (0.29; 0.90) |
| **Major Response** | 0.79 (0.59; 1.07) | 0.47 (0.15; 0.55) | 0.19 (0.09; 0.36) | |
| **Lymph node** | 0.90 (0.63; 1.30) | 0.35 (0.20; 0.82) | 0.54 (0.23; 0.95) | 0.65 (0.58; 0.68) |
| **Bone** | 1.28 (0.87; 1.63) | 0.83 (0.68; 1.23) | 0.90 (0.86; 1.19) | |

Absolute values of K$^{trans}$ (min$^{-1}$) reported with medians and interquartile ranges (IQR) between 25th and 75th percentile during therapy with Erlotinib and Bevacizumab.

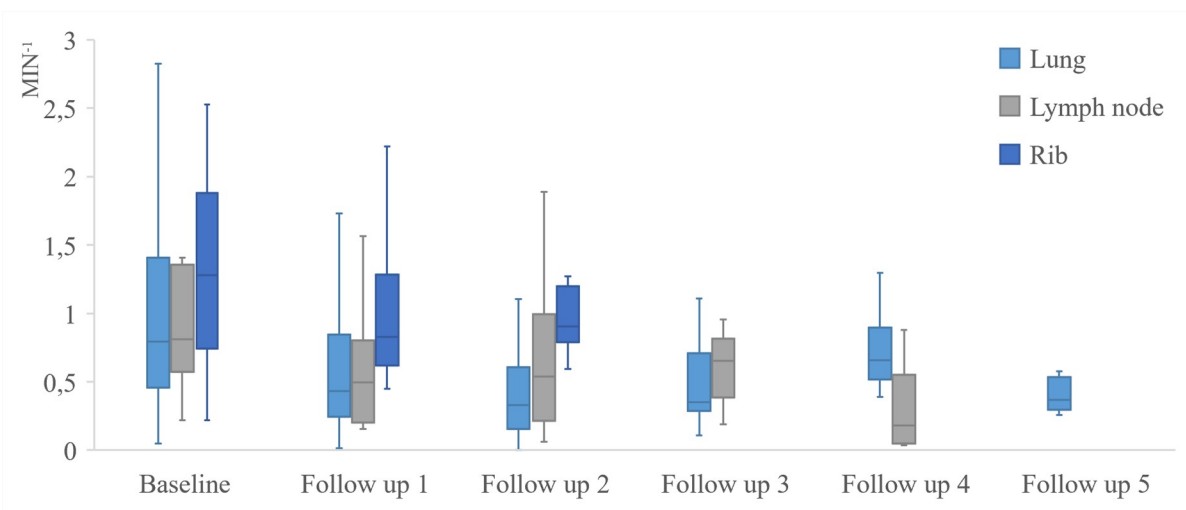

**Fig 3. Distribution of K^trans values.** Boxplot showing the median, the 25th and 75th percentiles respectively the 1.5 interquartile range below the 25th and above the 75th percentile of $K^{trans}$ (min$^{-1}$) during therapy with Erlotinib and Bevacizumab.

### Interrater agreement

The volume transfer coefficient $K^{trans}$ showed a moderate to good reliability in overall measurements with an ICC of 0.784 (95% CI: 0.716–0.841) and a wCV of 34.3% (95% CI: 30.3–38.4). $K^{trans}$ in pulmonary tumors $\geq$ 3 cm showed a good to excellent reliability with an ICC of 0.851 (95% CI: 0.773–0.907) and a wCV of 26.2% (95% CI: 23.0–29.4) while $K^{trans}$ in tumors < 3 cm showed a moderate to good reliability with an ICC of 0.813 (95% CI: 0.710–0.889) and a wCV of 40.2% (95% CI: 31.6–48.7). $K^{trans}$ in intrapulmonary tumors showed a good reliability with an ICC of 0.823 (95% CI: 0.761–0.873) and a wCV of 33.2% (95% CI: 28.9–37.5) while $K^{trans}$ in extrapulmonary tumors showed a poor to moderate reliability with an ICC of 0.335 (95% CI: 0,018–0,680) and a wCV of 39.9% (95% CI: 28.1–51.8) (see Table 3).

### Intrarater agreement

The pharmacokinetic parameter $K^{trans}$ showed a moderate to good reliability in overall measurements with a mean ICC of 0.713 (95% CI: 0.607–0.795) and a mean wCV of 27.5% (95%

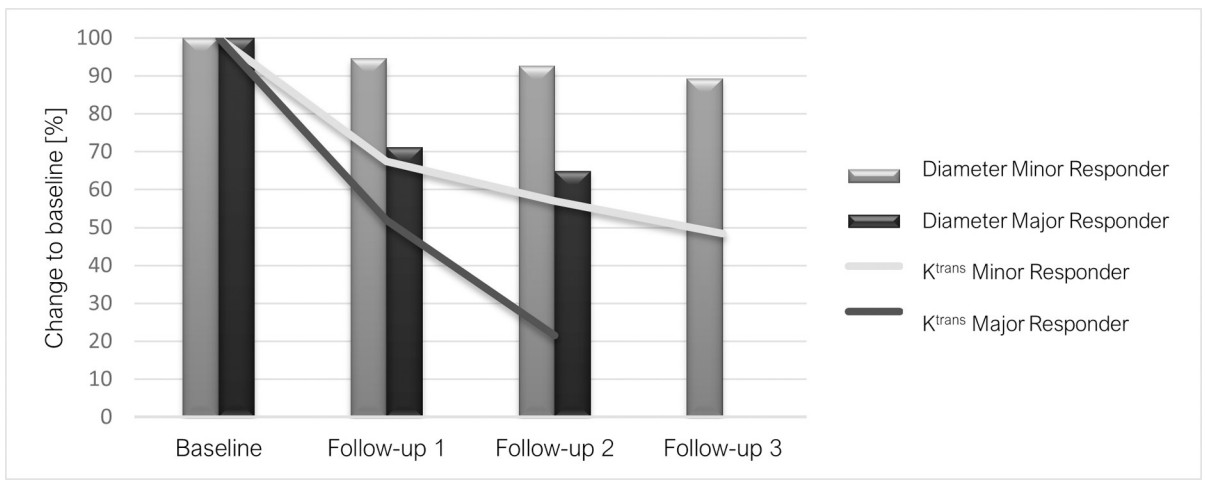

**Fig 4. Percentual change of K^trans and tumor size in minor and major responders during therapy with Erlotinib and Bevacizumab.**

**Table 3. Interrater reliability.**

| | | ICC (95%) | %wCV (95%) |
|---|---|---|---|
| **Overall** | | 0.784 (0.716; 0.841) | 34.3 (30.3; 38.4) |
| **Intrapulmonary** | *All sizes* | 0.823 (0.761; 0.873) | 33.2 (28.9; 37.5) |
| | *1–3 cm* | 0.813 (0.710; 0.889) | 40.2 (31.6; 48.7) |
| | *≥ 3 cm* | 0.851 (0.773; 0.907) | 26.2 (23.0; 29.4) |
| **Extrapulmonary** | | 0.335 (0.018; 0.680) | 39.9 (28.1; 51.8) |

Interrater reliability of $K^{trans}$ among 3 readers expressed as the interclass coefficient (ICC) and the within subject coefficient of variation (wCV), depending on tumor location and size.

CI: 24.6–30.4). Best intrarater agreement with good to excellent reliability was revealed in pulmonary tumors ≥ 3 cm with an ICC of 0.852 (95% CI: 0.758–0.912) and a wCV of 21.4% (95% CI: 19.0–23.8) (see Table 4).

## Discussion

Quantitative imaging has gradually become an important tool in oncological care. A broad number of published studies in solid cancer outside the lung have shown that DCE-MRI may provide important predictive and prognostic biomarker and allows early non-invasive treatment monitoring in various solid cancers [13–19]. However, the standardization of DCE-MRI and the use of (semi-)quantitative pharmacokinetic models is still an issue for the implementation of DCE-MRI in clinical routine. This limitation has been addressed by the Quantitative Imaging Biomarkers Alliance (QIBA) of the Radiological Society of North America (RSNA) [20] with the main claim of a high intra- and interrater agreement of the DCE-MRI and post-processing derived pharmacokinetic biomarkers, which is indispensable for the clinical implementation [21]. Thus, the purpose of this sub study was to evaluate the robustness of DCE-MRI in lung cancer.

The results of this study showed a moderate to good overall interrater reliability in patients with lung cancer. Especially in larger intrapulmonary lesions the DCE-MRI revealed a good to excellent interrater agreement. To our best knowledge, only an extremely limited number of studies of DCE-MRI derived pharmacokinetic parameters in lung tumors have been reported so far. In a study of van den Boogaart et al. 21 patients underwent paired DCE-MRI. In this study the interrater reproducibility of $K^{trans}$ in lung tumors measured by two readers was reported even higher with an ICC of 0.930 compared to 0.823 (95% CI: 0.761–0.873) in our study. The intrarater reliability was also calculated higher with an ICC of 0.984 against 0.713 in our study [22]. The better reliability reported by van den Boogaart et al. might be related to the

**Table 4. Intrarater agreement.**

| | | *Mean ICC* | *Mean wCV* |
|---|---|---|---|
| **Overall** | | 0.713 (0.607; 0.795) | 27.5 (24.6; 30.4) |
| **Intrapulmonary** | *All sizes* | 0.729 (0.618; 0.812) | 26.2 (23.2; 29,2) |
| | *1–3 cm* | 0.687 (0.487; 0.820) | 31.2 (25.2; 37.1) |
| | *≥ 3 cm* | 0.852 (0.758; 0.912) | 21.4 (19.0; 23.8) |
| **Extrapulmonary** | | 0.415 (0.000; 0.761) | 32.6 (24.3; 40.9) |

Mean Intraclass agreement of $K^{trans}$ in 2 readers, expressed as the intraclass coefficient (ICC) and the within subject coefficient of variation (wCV), depending on tumor location and size.

different ROI-placement and tumor annotation. First, in the previous study ROIs were drawn in only one single central slice, concretely defined by the largest diameter of the tumor according to RECIST while measurement in our study were performed in multiple slices which leaves more scope for ROI positioning and probably harms the reproducibility. However, since a single layer does not reflect the entire tumor microenvironment including necrotic areas, our method appears to reflect the more accurate $K^{trans}$ value reflecting the entire tumorous heterogeneity. Second, the tumor contouring in baseline and follow-up examinations was accomplished in one single session while the readers in our study were blinded to the further course of disease. However, our goal was to reflect everyday clinical practice as realistically as possible.

Nonetheless, the ICC obtained in our study was similar compared to other DCE-MRI studies in varying solid cancer and tumor locations in which reproducibility of DCE-MRI derived biomarkers has been reported. For example, Heye et al. reported $K^{trans}$ results in uterine fibroids which were manually measured by five readers with an interobserver ICC of 0.79 (95% CI: 0.70–0.85). Intrarater reproducibility was calculated with an overall ICC of 0.86 (95% CI: 0.77–0.92) [23]. In another study by Wang et al. where 21 patients with renal cell carcinoma underwent paired DCE-MRI the interobserver ICC for three readers was 0.686 (95% CI: 0.212–0.898) [9].

The QIBA DCE-MRI Technical Committee released the first version of a DCE-MRI Profile with the goal of better standardization of DCE-MRI acquisitions. The committee proposed a within-subject coefficient of variation of 20% for the quantitative transfer constant $K^{trans}$ in respect for a clinical interpretation of the specific $K^{trans}$ measurement and the potential implication for clinical decision making for solid tumors with at least 2 cm in diameter. However, the claim of a coefficient of variation of 20% was made of the available literature of DCE-MRI in various solid cancers. This suggested that a change of 40% might be required in a single patient to be considered as a real treatment effect. But these papers did not take specific account of the wCV within lung tumors which was the dedicated focus of our work. In addition, the limit of the coefficient of variation was determined for test-retest data, not for interobserver reliability. We revealed interobserver variations for intrapulmonary lesions overall from 28.9 to 37.5% and in smaller lesion of 1–3 cm from 31.6 to 48.7%. Our results suggest that a wCV of 20% and a change of more than 40% for treatment response may not be optimal in lung tumors. The knowledge of this technical limitation of DCE-MRI for quantitative measurements of $K^{trans}$ in lung cancer is particularly important, but even more important for clinical decision making and for the definition of possible thresholds for response classification. Thus, uniform threshold of 40% according to the QIBA for all tumors, or the RECIST thresholds for partial response by decrease in the sum of size $\leq$ 30%, or progressive disease $\geq$ 20% would not be within our limits of $K^{trans}$ in lung tumors. Based on our results a threshold of below or above 50% for partial response or progressive disease for $K^{trans}$ seems to be more reasonable but needs further confirmations by larger clinical studies. Moreover, based on our results a minimum lesion diameter of lung cancer in DCE-MRI of greater than 1 cm as defined measurable target lesion by RECIST, or 2 cm as mentioned of the QIBA must be evaluated.

In our study we observed a poor reproducibility of $K^{trans}$ measurements in extrapulmonary thoracic lesions including osteolytic bone lesions and lymph node metastasis. This could be caused by the relatively small amount of extrapulmonary lesions analysed and a poor visualization due to an unscheduled MRI protocol. However, another reason could be the higher rate of artefacts due to greater movements of the ribcage, pulsation of the aorta and heartbeat within the mediastinum, as well as the smaller lesion size. Soon, recent advances in image acquisition techniques like compressed sensing could bring about a significant improvement in image quality and reliability.

There are several limitations in this study that need to be addressed. First, we assumed an amount of three readers of different expertise. The reproducibility could differ from results in case measurements of only experienced radiologists in DCE-MRI. However, we wanted to represent the real clinical scenario. Second, the study only reports the reproducibility of $K^{trans}$ as a single pharmacokinetic biomarker derived from DCE-MRI. Other semiquantitative biomarkers, such as the blood-normalized initial-area-under-the-gadolinium-concentration curve ($IAUGC_{BN}$), time to peak, wash-in and wash-out rate, or quantitative biomarkers, such as the efflux rate constant ($K_{ep}$) are known to reflect also tumor perfusion as well as microvasculature and permeability and all these may demonstrate different quantitative results [2,6,9,15,23–27]. Nevertheless, we decided to evaluate $K^{trans}$ as the most used and the most established pharmacodynamic biomarker in DCE-MRI [28,29]. Third, the study focused on non-small cell lung carcinoma and its thoracic metastases. Since it is known that different tumor entities have different physiological characteristics that can be exerted on their microvasculature, it is not known how our results are transferable to other cancer entities within the lung, pulmonary metastases or even to different types of lung cancer. Forth, the lesions were painstakingly evaluated manually. Although the manual, multi-sliced evaluation is intended to reflect better the tumor heterogeneity, but it is impractical due to the high expenditure of time in clinical practise [6]. The time-saving single-slice ROI would presumably not reflect micro-vascularization and membrane permeability of the whole tumor and would not reflect the tumor heterogeneity in the same manner. The solution could be an automatic measurement of the mean and maximum value of $K^{trans}$, similar with the maximal standardized uptake value ($SUV_{max}$) in PET-CT. However, for DCE-MRI to establish itself as a reliable therapy monitoring procedure in everyday clinical practice, it is essential to maintain reproducibility at the highest possible level. Ideally, patients should be examined during their follow-up by a single reader in one session and in the same centre. A standardized procedure should be established for the scan and for the subsequent evaluation, leaving as little scope as possible for the placement of the ROIs. Thus, the standardization of the DCE-MRI acquisition is one of the main goals of the QIBA initiative of the RSNA and described in the published profile [30]. Whether a semi-automated or even fully automated evaluation will prove itself in the future remains to be explored.

In summary, with $K^{trans}$, DCE MRI offers a reliable quantitative biomarker for early non-invasive therapy assessment in lung cancer patients but with a coefficient of variation of up to 48.7% in smaller lung lesions.

## Supporting information

**S1 Dataset. $K^{trans}$ raw data.** $K^{trans}$ values from all readers for all lesions listed by location and size used for the calculation of ICC and wCOV.
(XLSX)

## Author Contributions

**Conceptualization:** Jean-Philip Daniel Weber, Judith Eva Spiro, Matthias Scheffler, Jürgen Wolf, Lucia Nogova, Marc Tittgemeyer, David Maintz, Hendrik Laue, Thorsten Persigehl.

**Data curation:** Jean-Philip Daniel Weber, Judith Eva Spiro, Hendrik Laue, Thorsten Persigehl.

**Formal analysis:** Hendrik Laue, Thorsten Persigehl.

**Investigation:** Jean-Philip Daniel Weber, Judith Eva Spiro, Matthias Scheffler, Jürgen Wolf, Lucia Nogova, Marc Tittgemeyer, Hendrik Laue, Thorsten Persigehl.

**Methodology:** Jürgen Wolf, Thorsten Persigehl.

**Project administration:** Matthias Scheffler, Jürgen Wolf, Hendrik Laue, Thorsten Persigehl.

**Resources:** Lucia Nogova, Thorsten Persigehl.

**Software:** Hendrik Laue, Thorsten Persigehl.

**Supervision:** Matthias Scheffler, Jürgen Wolf, David Maintz, Thorsten Persigehl.

**Visualization:** Hendrik Laue, Thorsten Persigehl.

**Writing – original draft:** Jean-Philip Daniel Weber.

**Writing – review & editing:** Thorsten Persigehl.

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
