## [Decision Letter · Decision Letter 0]

10 Jan 2022

PONE-D-21-31257Reproducibility of dynamic contrast enhanced MRI derived transfer coefficient Ktrans 5 in lung cancerPLOS ONE

Dear Dr. Weber,

Thank you for submitting your manuscript to PLOS ONE. After careful consideration, we feel that it has merit but does not fully meet PLOS ONE’s publication criteria as it currently stands. Therefore, we invite you to submit a revised version of the manuscript that addresses the points raised during the review process.

We look forward to receiving your revised manuscript.

Kind regards,

Pascal A. T. Baltzer, M.D.

Academic Editor

PLOS ONE

Journal Requirements:

Reviewers' comments:

Reviewer's Responses to Questions

**Comments to the Author**

1. Is the manuscript technically sound, and do the data support the conclusions?

Reviewer #1: Yes

2. Has the statistical analysis been performed appropriately and rigorously? 

Reviewer #1: Yes

3. Have the authors made all data underlying the findings in their manuscript fully available?

Reviewer #1: No

4. Is the manuscript presented in an intelligible fashion and written in standard English?

Reviewer #1: Yes

5. Review Comments to the Author

Reviewer #1: In this study, the authors examined intra- and interreader variability of Ktrans, a DCE-MRI based quantitative biomarker of tissue perfusion and vascular permeability, in solid lung tumors. Ktrans showed a moderate to good reliability overall, which correlated with tumor size. The manuscript is well written and the technical and analytical approaches are sound. I only have a few minor suggestions which should be adressed prior to publication.

1. The abstract conveys the most important information in a very comprehensible manner. There is a minor grammatical mistake in the last sentence, as it says „DCE MRI offers with Ktrans a reliable quantitative biomarker“, which should rather read „With Ktrans, DCE MRI offers a reliable quantitative biomarker“, i.e. the words should be repositioned.

2. The authors should look into the distributions of Ktrans values around their means per timepoint, specifically their deviations from normality, since the calculation of confidence intervals for ICCs assumes a normal distribution. Since Ktrans values are zero-bound, a truly normal distribution is not possible. For Fig. 3, a graphical representation of medians (i.e. a box plot in the style of Tukey) should be chosen, rather than means and SDs.

3. The dataset is quite heterogenous, as it consists of both extra- and intrapulmonary lesions with variable sizes. Due to the low number of intrapulmonary lesions, reliability measures on these might not be conclusive. The authors might consider to exclude these lesions from their analyses. Alternatively, for table 3 and 4, lesions could be grouped according to both size and location, i.e. showing data for for intrapulmonary lesions larger than 3 cm, extrapulmonary lesions larger than 3 cm, intrapulmonary lesions smaller than 3 cm and extrapulmonary lesions smaller than 3 cm.

4. Table 4 is titled „Intraclass agreement“, whereas the corresponding paragraph in the text is titled „Intrarater agreement“. The authors should change the title of the table to avoid confusion.

5. In the dicussion, the authors claim that Ktrans is „probably the most used and most established DCE-MRI biomarker“, to justify why they chose this parameter over other semiquantitative biomarkers. Is this claim based on purely experience, or if there literature available?

6. This might not concern the authors directly, but I was not forwarded the supporting information and can therefor not validate that all data is fully available. For this reason I chose „No“ in the corresponding review box.

6. PLOS authors have the option to publish the peer review history of their article (what does this mean?). If published, this will include your full peer review and any attached files.

Reviewer #1: **Yes: **Martin L. Watzenböck

---

## [Author Response · Author response to Decision Letter 0]

8 Feb 2022

Responses to Review Comments

1. We apologize for the grammatical mistake. The sentence structure has been modified in the revised manuscript according to your suggestion. The correction was made both in the abstract on page 2, line 41 and in the discussion on page 16, line 340.

2. Thank you for the legitimate suggestion for improvement. In both Table 2 and Fig 3, due to the heterogenous distribution, the means and standard deviations for Ktrans were misleading. Table 2 now shows the median values of Ktrans and the interquartile ranges (IQR). Fig 3 has also been renewed and now shows the Ktrans values in form of boxplots in the style of Tukey.

3. We thank you for the concrete and legitimate proposals. The division of the lesions and the resulting structure of our work was subject of intense debates in our research group. In fact, the extrapulmonary metastases are small in number, which reduces statistical significance. This point is therefore explicitly taken up again in the discussion section on page 14, line 302-309. Nevertheless, we did not want to undermine the poor reproducibility of extrapulmonary metastases, with the aim of pointing out that future measurements of such lesions may require adjustments to protocols and study settings.

Furthermore, we also considered subdividing the extrapulmonary lesions according to their size. In contrast to the intrapulmonary lesions, however, there does not seem to be a meaningful cut-off analogous to the T stage or the RECIST criteria. This would make any subdivision look arbitrary. Since in our dataset the lymph nodes are already rather small (2.6 ± 0.5 cm) and the bone metastases are rather larger (4.3 ± 0.8 cm), a natural split is already given here in a certain respect.

While we fully understand your concerns, we hope this provides some explanation of our perspective to illustrate why we want to keep the extrapulmonary lesion as part of our work and refrain from making any further subdivisions.

4. Thank you for the kind notice. To avoid possible confusion, it is important to us to use a uniform vocabulary. The designation was therefore adjusted according to your suggestion.

5. We acknowledge that this was a vague wording. References 28 (Di N et al.) and 29 (Jiang T et al.) have been added to support our statement. The corresponding changes were added to the reference list.

6. According to the PLOS Data Policy the full dataset of Ktrans values underlying the statistical ICC and wCOV calculation has been added to the Supporting information named “S1_Dataset”. The dataset contains Ktrans values measured by every single reader of all 107 tumor lesions sorted by location and size.

---

## [Editor Report · Decision Letter 1]

23 Feb 2022

Reproducibility of dynamic contrast enhanced MRI derived transfer coefficient Ktrans in lung cancer

PONE-D-21-31257R1

Dear Dr. Weber,

We’re pleased to inform you that your manuscript has been judged scientifically suitable for publication and will be formally accepted for publication once it meets all outstanding technical requirements.

Kind regards,

Pascal A. T. Baltzer, M.D.

Academic Editor

PLOS ONE

Additional Editor Comments (optional):

I have reviewed your revised manuscript and found the changes to my full satisfaction. This paper is a worthy contribution to its field. Please excuse my apology for the long review process that was entirely due to me being overloaded by work - next time I will be faster...
---

## [Editor Report · Acceptance letter]

28 Feb 2022

PONE-D-21-31257R1 

Reproducibility of dynamic contrast enhanced MRI derived transfer coefficient K^trans^ in lung cancer 

Dear Dr. Weber:

I'm pleased to inform you that your manuscript has been deemed suitable for publication in PLOS ONE. Congratulations! Your manuscript is now with our production department. 

Kind regards, 

on behalf of

Dr. Pascal A. T. Baltzer 

Academic Editor

PLOS ONE